# Model-Driven Engineering Techniques and Tools for Machine Learning-Enabled IoT Applications: A Scoping Review

**DOI:** 10.3390/s23031458

**Published:** 2023-01-28

**Authors:** Zahra Mardani Korani, Armin Moin, Alberto Rodrigues da Silva, João Carlos Ferreira

**Affiliations:** 1ISCTE, Instituto Universitário de Lisboa (ISCTE-IUL), ISTAR, 1649-026 Lisbon, Portugal; 2Hydraulics and Environment Department, LNEC, 1700-066 Lisbon, Portugal; 3School of Computation, Information, and Technology (CIT), Technical University of Munich, D-80333 Munich, Germany; 4INESC-ID, Instituto Superior Técnico, Universidade de Lisboa, 1000-029 Lisbon, Portugal; 5Inov - Inesc Inovação—Instituto de Novas Tecnologias, 1000-029 Lisbon, Portugal

**Keywords:** model-driven engineering, internet of things, data analytics and machine learning, time series, literature review, scoping review

## Abstract

This paper reviews the literature on model-driven engineering (MDE) tools and languages for the internet of things (IoT). Due to the abundance of big data in the IoT, data analytics and machine learning (DAML) techniques play a key role in providing smart IoT applications. In particular, since a significant portion of the IoT data is sequential time series data, such as sensor data, time series analysis techniques are required. Therefore, IoT modeling languages and tools are expected to support DAML methods, including time series analysis techniques, out of the box. In this paper, we study and classify prior work in the literature through the mentioned lens and following the scoping review approach. Hence, the key underlying research questions are what MDE approaches, tools, and languages have been proposed and which ones have supported DAML techniques at the modeling level and in the scope of smart IoT services.

## 1. Introduction

Internet of things (IoT) generally refers to many cyber-physical things, including objects and devices around us that are connected to the Internet and can be controlled, accessed, and managed by a myriad of applications running on smartphones and other computational environments [1]. Things can interact with each other and with the users of the digital services that are based on the IoT, thus, called IoT applications or services [2]. Sensors and devices are connected to a network through which they can interact with each other and their users. The IoT concept can be illustrated with a simple example, such as connecting a smartphone to a TV, or with more complex examples, such as monitoring urban infrastructures and traffic [3,4]. IoT-based applications monitor, control, or interact with the physical world. The most popular IoT application domains are healthcare, environment, smart cities, commercial, and industrial [2]. Furthermore, artificial intelligence techniques are being used in IoT applications to add intelligence to devices and related services [5].

IoT data analytics is the process of analyzing the data that are produced from IoT devices and services [6]. In this scope, data analytics and machine learning (DAML) techniques help to make IoT services smarter by finding patterns, hidden correlations, trends, inferences, and actionable insights that have not been seen before. This would lead to more intelligence, better decision-making, performance, automation, productivity, and accuracy. However, the huge increase in the amount and complexity of data poses new challenges for data analytics [7]. In particular, non-i.i.d data (i.e., data instances for which the usual assumption of independent and identically distributed data may not hold), such as sequential time series data requires reliable methods and techniques that should be supported in tools that are focused on IoT data analytics.

However, designing and developing IoT applications requires skills in various areas. Furthermore, several technologies, such as programming embedded systems, communication, networks, data science, and web engineering [8]. To successfully implement IoT applications that are often quite complex, efficient software development methodologies, approaches, and solutions have been provided to address various issues, including heterogeneity, collaborative development, reusability of software artifacts, and self-adaptation [9,10].

One such proposal is model-driven engineering (MDE), which is a general engineering approach that focuses on using models as the primary artifact for software development [10]. In MDE, models can be used to capture the structure and behavior of a system, and these models can then be transformed into the final implementation of the system using model transformations and code generators. MDE aims to improve the efficiency of the software development process by automating the transformation of models into code, and by providing domain-specific abstractions and notation that make it easier to reason about and design complex systems. MDE is often used with domain-specific languages (DSLs), as the models used in MDE are often expressed using these languages. DSLs are languages designed for a particular domain, such as a particular problem or a particular domain of expertise [11]. DSLs are designed to be more concise and more accessible to use than general-purpose languages; they often provide domain-specific abstractions and notation that make it easier to express solutions to problems in that domain [12].

According to the MDE paradigm, a key artifact is the “model”; thus, the whole software development cycle revolves around models. Many MDE approaches aim to generate code for various platforms automatically [10]. An MDE approach can follow two general stages. The first stage involves the creation of MDE artifacts such as languages and tool workbenches with features like authoring, model-to-model, and model-to-text transformations. In the second stage, people without advanced computer training can easily use these artefacts to design and develop their software systems [12].

However, because various MDE tools and languages are emerging, understanding and selecting the most suitable tools and languages are always challenging, especially for new developers or researchers in the area. From our knowledge, there is no general framework or guideline to help to select which tool(s) and language(s) should be used for supporting the various MDE activities in the development of IoT applications (or just “MDE4IoT” for short) [13,14].

Furthermore, the current literature that reviews MDE4IoT does not provide a comprehensive and complete classification and analysis of the languages, model-based transformations, application domains, and even the supported DAML techniques, as we discuss in this paper. For example, Salman et al. [15] analyzed only a few textual and graphic modeling languages and respective tools, but without an accurate classification, and they did not classify those tools and languages or identify their specific features like model-to-model or model-to-text transformations, as we propose in this paper. A further discussion of the related work is presented below in Section 3.

This paper presents a scoping review of research studies that have proposed or used MDE4IoT. This review includes an accurate classification of the used modeling languages, model-to-model, and model-to-text transformations, but also on identifying the application domains and even on the DAML techniques with a particular focus on time series data. This review can assist researchers and developers in selecting the best MDE tools, languages, and techniques for their IoT projects. Moreover, this paper provides a general discussion of future trends by arguing that more work on DAML, in the context of MDE4IoT, is a worthy research area. We select, classify and review 68 studies presenting the stat-of-the-art in MDE4IoT, a few of them focusing on DAML techniques. In addition, we identify the limitations of prior work and discuss some challenges that address these shortcomings.

This paper is organized into seven sections. Section 2 introduces the background by reviewing the basic concepts of MDE, IoT, and DAML. Section 3 presents the related work, i.e., other papers that have also addressed literature reviews on similar topics. Section 4 describes the followed research methodology based on the scoping review approach. Section 5 presents the results from the scoping review. Section 6 discusses the results obtained from the review and identifies the limitations. Finally, Section 7 presents the main conclusion and open issues.

## 2. Background

The key concepts considered in the scope of this research are MDE, IoT, and DAML, as introduced below.

### 2.1. Model-Driven Engineering (MDE)

Model-driven engineering (MDE) is a software development approach that emphasizes using models as a primary means for specifying, designing, and implementing software systems. A simple and popular formula used to describe MDE is the following [16]: Models + Transformations = Software, thus, we briefly introduce these essential concepts as follows [14,16,17]:

A *model* is an abstract representation of a system that eliminates unnecessary details to increase the users’ understanding of the system under consideration. In addition, a metamodel is a particular model that describes the structure of a modeling language, i.e., a metamodel defines the abstract syntax of a modeling language [14,17]. *Transformations* are special programs that convert models to other models or to text (e.g., source code) according to specific rules previously defined. Model transformations are commonly classified as model-to-model (M2M) and model-to-text (M2T). In M2M, both the input and the output of the transformation are models, whereas in M2T, the input is a model, but the output is text, such as source code in a programming language like Java or Python.

MDE involves creating and manipulating abstract, high-level models that represent the different aspects of a system, such as its functionality, structure, and deployment. These models can then be analyzed, validated, and transformed using various tools and techniques to generate the actual software code and related artifacts [14,18]. A common goal of MDE is to improve the efficiency and reliability of software development by providing a more abstract and flexible way of specifying and designing systems. MDE also aims to reduce the gap between the conceptual models and the implementation by automating the transformation and code generation process. MDE is often used in complex, large-scale software projects where traditional development approaches can be very cumbersome, and error-prone [19]. MDE relies on using modeling languages, such as UML, SysML, and domain-specific languages (DSLs); and model-driven tools, such as model checkers, and M2M and M2T transformations. MDE is commonly specialized by more concrete approaches, such as model-driven architecture (MDA), model-driven development (MDD) [14], model-based testing (MBT) [20].

### 2.2. Internet of Things (IoT)

The Internet of things (IoT) refers to the interconnected network of physical devices, vehicles, buildings, and other objects embedded with sensors, software, and connectivity, allowing them to collect and exchange data over the Internet. These connected devices can range from smart appliances, and wearable devices to industrial equipment and infrastructure [21,22].

The main goal of IoT is to improve the efficiency, productivity, and safety of various industries and applications by enabling the seamless exchange of data and information between devices and systems. IoT also enables the creation of new services and business models by leveraging data analytics and machine learning techniques to extract insights and value from the collected data.

IoT relies on various technologies, such as sensors, communication protocols, and cloud computing, to enable the connectivity and interoperability of devices. It also involves using various frameworks, such as the IoT Reference Model, to define the architecture and interoperability of IoT systems.

IoT has the potential to transform various sectors, such as transport and mobility, healthcare, manufacturing, and agriculture, by providing real-time data and insights that can improve the decision-making process and optimize the performance of various systems and processes [21,23]. However, IoT also raises concerns about privacy, security, and sustainability, as the proliferation of connected devices can create new vulnerabilities and risks. Therefore, developing and deploying IoT applications requires careful consideration of these issues and adopting appropriate security measures and best practices.

### 2.3. Data Analytics and Machine Learning (DAML)

Data analytics and machine learning (DAML) refers to the use of techniques to extract insights and knowledge from data. Data analytics (DA) involves the use of statistical, computational, and visualization techniques to analyze and interpret data. DA can be used to identify patterns, trends, and relationships in data, and to make predictions and decisions based on this analysis. On the other hand, machine learning (ML) involves the use of algorithms and models that can learn and improve their performance over time without explicit programming. This is achieved by training the algorithms on data and allowing them to automatically discover patterns and relationships in the data.

Together, DAML can be used to solve a wide range of problems, including predictive modeling, classification, clustering, and optimization [24]. It can be applied to a variety of domains. These approaches are broadly classified into four types [25]: supervised, unsupervised, semi-supervised, and reinforcement learning.

Most of the data that IoT devices produce are sequential time series data. This means the order of data instances matters, and instances are not independent. The main goal of time series analysis is to *learn* patterns in this kind of data to find valuable features, predict future patterns, and find out how different data streams are related. Auto-correlation is a property of time series data. This means that the current value in the time series is linked to values from the past. In linear models, the current value is a linear function of the previous values. In nonlinear models, the current value is not a linear function of the previous values. If the properties of a stochastic process change over time, it is hard to predict its future values based on what has happened in the past, and this is called a non-stationary process. Auto-Regressive Integrated Moving Average (ARIMA), Hidden Markov Models (HMMs), and Recurrent Neural Networks (RNNs) are some of the well-established methods to model time series data [26].

## 3. Related Work

Since this research reviews the state-of-the-art of MDE tools and languages for IoT, this section introduces other related secondary studies, i.e., surveys, systematic mapping, and systematic review articles. Table 1 presents a summary of these related works.

Sabin et al. [27] analyzed the state of the art in MDE4IoT by reviewing 26 papers using Google Scholar and the snowballing method. These authors investigated MDE techniques for addressing IoT challenges. Furthermore, they examined IoT concepts related to MDE4IoT approaches, which included: device description, discovery, deployment, self-adaptation, service composition, cloud computing, fog, edge computing, and middle-ware frameworks. The most addressed IoT concept in this review was service composition, which was identified in 16 studies reviewed.

Felicien et al. [28] showed the current state of existing low-code platforms for developing IoT systems. They analyzed 16 platforms and defined a set of features that corresponded to the functions and services that each platform supports. In particular, they focused on languages and tools available in the MDE field and the emergent low-code development platforms for the IoT domain.

Muzaffar et al. [29] discussed the use of MDE tools in cloud computing scenarios. They defined research questions regarding MDE paradigms and techniques applied in cloud computing (e.g., SaaS, PaaS, and IaaS), and the leading cloud computing tools that include some MDE features.

Abshir Mohamed et al. [30] reviewed MDE techniques, tools, and languages used in cyber-physical systems (CPSs) and IoT components that are modeled. They raised four research questions, and to answer them, they analyzed 140 papers published between 2010–2018. The domain of their study was CPS, while ours is IoT services with a focus on DAML techniques.

Casalaro et al. [31] provided a framework for the software engineering research in MDE for the particular class of mobile robotic systems (MRS). That framework includes the following aspects: the types of robots supported by existing MDE approaches; the types and characteristics of MRS that are engineered using MDE approaches; description of how MDE approaches support the engineering of MRS; how existing MDE approaches are validated; and how tools support the existing MDE approaches.

Mashkoor et al. [32] presented a systematic mapping study on MDE in the context of safety and security systems. They analyzed key questions such as what are the commonly used methods and tools in this field, their development phases, and the most common application domains that have been evaluated?

Salman et al. [15] performed a systematic literature review on DSLs for the IoT by considering four research questions: what are the most common hardware and software platforms in each domain; what are the used DSLs; how are these DSLs evaluated; and, what are the scenarios in which these DSLs were applied.

Edsonde et al. [33] selected and analyzed 63 papers focusing on DSLs and concluded that robot self-adaptation is a well-established area. Still, with the advancements in machine learning techniques, it is possible to employ model-based approaches to make robotic software even more autonomous.

Differently, our review focuses on identifying and classifying the MDE tools and languages used to develop IoT services for different application domains. This review includes analyzing the involved tools, types of model-based transformations (e.g., M2M and M2T), the transformation-supported tools used, and even the DAML techniques provided by these tools.

## 4. Research Methodology

There are several MDE tools and languages for developing IoT applications; however, selecting the most suitable options is always challenging, especially for new developers or researchers. From our knowledge, there is no general framework or guideline to help select which tool(s) and language(s) should support MDE4IoT activities. For instance, the current reviews on MDE4IoT do not provide a comprehensive and complete classification and analysis of the languages, model-based transformations, application domains, and even DAML techniques, as proposed in this review.

This paper presents a scoping review of research studies that have proposed or used MDE technologies for IoT applications. A *scoping review* is a type of literature review used to map the key concepts, types of studies, and sources of information within a specific research area [34,35]. It is typically used to identify the breadth and depth of existing research on a particular topic and to identify gaps in the literature that may need to be addressed in future studies. The scoping review method involves a systematic search and analysis of the literature, using predefined inclusion and exclusion criteria to identify relevant studies. The findings of a scoping review are usually presented in a narrative or tabular format and can be used to inform the design and implementation of future research studies.

A scoping review and a systematic literature review (SLR) are both types of literature reviews, but they have some key differences in terms of purpose, methodology, and output [35,36]: *Purpose*. The main purpose of a scoping review is to map the key concepts, types of studies, and sources of information within a specific research area. It is used to identify the breadth and depth of existing research on a particular topic and to identify gaps in the literature that may need to be addressed in future studies. On the other hand, the main purpose of an SLR is to summarize and critically evaluate the existing evidence on a specific research question or topic. *Methodology*. A scoping review typically has a less rigorous methodology than an SLR. Scoping reviews often use more inclusive inclusion criteria, and the search and analysis of the literature are typically less formal and less comprehensive. On the other hand, SLRs follow a more rigorous methodology, including a comprehensive search strategy, predefined inclusion and exclusion criteria, and a formal data extraction and synthesis process. *Output*. The output of a scoping review is typically presented in a narrative or tabular format and can be used to inform the design and implementation of future research studies. On the other hand, the output of an SLR is typically presented in the form of a systematic review or a meta-analysis, which summarizes and critically evaluates the existing evidence on a specific research question or topic. In summary, a scoping review is a broad overview of the existing literature on a topic and is used to identify gaps and opportunities for future research, while an SLR is a more in-depth and rigorous examination of the existing literature on a specific research question or topic, to summarize and critically evaluate that topic.

A scoping review can follow a 5-stage process, as originally discussed by Arksey and O’Malley [37], which is a rigorous process of transparency that allows replication of the search technique and enhances the dependability of the study’s conclusions. As shown in Figure 1, scoping review involves a sequence of the following five stages (or tasks): (1) Identify the initial research questions; (2) identify relevant studies; (3) select the relevant studies; (4) chart the data; and (5) summarize and report the results. We ran these stages, reacted to each stage, and repeated some stages, as needed, to ensure that all of the literature was covered and analyzed. This means that the process ran iterative (as suggested in the figure), i.e., ran several times until we got confident with the results.

The current section is organized around the presentation of these stages. Stages 1 to 4 are presented in the following subsections; Stage 5 is in Section 5.

### 4.1. Identify the Research Questions (Stage-1)

The first stage of the scoping review consisted of defining the initial research questions. These questions help frame and conduct the review and are the following:

**RQ1**. What is the current state of MDE languages and tools for developing IoT applications? This question is broken down into the following sub-questions: **RQ1.1**. What modeling languages are used? **RQ1.2**. What tools are used? **RQ1.3**. What model transformations and the respective supported languages and tools are employed? **RQ1.4**. What are the outputs of M2M and M2T transformations?

**RQ2**. What application domains are these MDE approaches mostly applied to?

**RQ3**. What DAML techniques are supported by these MDE approaches?

### 4.2. Identify Relevant Studies (Stage-2)

The second stage of the research aims to identify relevant studies in the field of MDE4IoT by conducting a comprehensive literature search. This stage consists of three activities: (1) defining the search expression or search terms, (2) discussing and defining the eligibility criteria, and (3) selecting databases or repositories to collect research papers.

**Search expression**: To obtain a broad coverage of the available literature, Arksey and O’Malley [37] recommend using a broad definition of keywords for search terms. Thus, key concepts and search terms were defined to capture most of the literature related to the MDE4IoT topic. Search tools and Boolean operators were used to narrow, broaden, and combine literature searches. The defined search expression for this study is *(“model-driven development” OR “model-driven engineering” OR “model-driven architecture” OR “model-based approach” OR “model-driven approach” OR “domain specific model*” OR “metamodel”) AND (“IoT*” OR “Internet of thing*”)*.

**Eligibility criteria**: To ensure the relevance of the selected papers, the following inclusion and exclusion criteria were defined:

*Inclusion criteria*: The study must propose at least one MDE tool, language, or technique for IoT; The study must be a peer-reviewed manuscript, i.e., a journal, conference, or workshop paper; The study must be published between 2010 and 2023.

*Exclusion criteria*: The study is a secondary manuscript, i.e., a survey, systematic mapping, or systematic review; The study is not applied or relevant to the IoT domains, or it does not follow an MDE approach; The study is a type of educational, editorial, tutorial, or other, i.e., it is not a scientific paper; The study is written in other languages than English.

**Databases of research papers**: To identify peer-reviewed literature, the following electronic databases were used: IEEE Explore (https://ieeexplore.ieee.org, accessed on 13 December 2022), ACM Digital Library (https://dl.acm.org, accessed on 13 December 2022), ScienceDirect (https://www.sciencedirect.com, accessed on 13 December 2022), Web of Science (https://clarivate.com, accessed on 13 December 2022), SCOPUS (https://www.scopus.com, accessed on 13 December 2022), and DBLP (https://dblp.org, accessed on 13 December 2022). These databases are considered the most popular and widely used in the IT domain.

### 4.3. Select Relevant Studies (Stage-3)

The third stage of the scoping review is the most critical and time-consuming stage of the research. This stage involves selecting the relevant studies that will be included in the review. Two research team members (ZM and AM) worked together to conduct an extensive selection of relevant studies. This stage was guided by the PRISMA framework [38] and involved the following activities:

**Identifying papers**: The research expression (previously defined) was used to search the selected databases, as well as Google Scholar (n = 621).

**Screening the title and abstract of the papers**: All the collected papers were screened based on their title and abstract, and duplicates were eliminated (n = 511). Papers that were secondary studies or not relevant to the MDE4IoT topic were also eliminated (n = 100).

**Assessing the full papers**: The remaining papers were assessed in full, and the eligibility criteria were applied to eliminate those papers that were not relevant (n = 68).

**Final set of included papers**: The final set was reached (n = 68).

Figure 2 summarizes the papers collected in each stage based on the PRISMA guideline [38]. This process of selecting relevant studies ensures that the review is comprehensive, relevant, and unbiased.

### 4.4. Chart the Data (Stage-4)

The fourth stage of the scoping review involves charting the selected papers by creating summaries of each paper, including information on the author, year, location of study, study design, study methods, and sample size, as well as a brief comment on the limitations and recommendations of the individual study.

We reviewed the final 68 papers to confirm that the selection was appropriate. Table A1 provides an overview of these papers, including their titles, years, publication types, countries, names of journals/conferences, and the number of authors.

In terms of **distribution of papers by year and publication type** (see Figure 3), we found that 40 studies were published in conferences, 19 in journals, and 9 in workshops. Researchers in this field prefer conferences over journals, except for 2022, which showed increased journal papers.

Regarding the **distribution of papers by year and the number of authors**, Figure 4 illustrates the authorship pattern for the past thirteen years. The study classified the number of authors in each paper into six clusters: single authors, two authors, three authors, four authors, five authors, and six or more authors. It was found that 25 papers were published by four authors, 22 papers by three authors, and 7 papers by five. Additionally, 6 papers were authored by two authors and 1 paper by a single author, indicating a growing trend of collaboration among researchers.

The authors’ affiliations were considered to determine the **country of origin for each publication**. As shown in Figure 5, the majority of the 68 papers were from Spain (11), Germany (9), France (8), Belgium (6), and Italy (5). Figure 5 shows the distribution of papers from the remaining 32 countries.

Regarding the **distribution of papers by year and journal**, 21 papers were published in 14 journals. Figure 6 displays the distribution of papers per journal from 2010 to 2022, showing an increase in journal publications in recent years.

Finally, Figure 7 illustrates the **distribution of papers by year of publication and digital databases**. It shows that 2017 and 2019 had the highest number of articles, with the majority of publications found in IEEExplore (36), followed by the ACM digital library (10), Science Direct (4), Web of Science (3), and other databases (14).

### 4.5. Summarize and Report the Results (Stage-5)

The results are summarized and reported in the fifth stage of the scoping review process, and these results are presented and discussed in Section 5.

## 5. Results Analysis

This section presents the results of the analysis of the responses to the research questions defined in Section 4.1.

### 5.1. What Is the Current State of MDE Languages and Tools for Developing IoT Applications? (RQ1)

The focus of this question is to analyze the state of the art on the MDE languages, tools, and model-based transformations used in developing IoT applications. However, this question is divided into the following sub-questions.


**What modeling languages are used? (RQ1.1)**


According to the MDE, the use of modeling languages helps define models with different perspectives (e.g., structure and behavior) and different levels of abstraction (e.g., platform-independent and platform-specific). These models support the design and development of complex software applications.

From the complete set of 68 reviewed studies, 29 studies reported the creation or the use of different DSLs (P1, P3, P5, P6, P7, P9, P10, P13, P14, P18, P19, P20, P24, P29, P30, P32, P35, P36, P39, P46, P41, P48, P50, P56, P59, P65, P66, P68); 11 studies use UML (P61, P55, P53, P51, P42, P38, P37, P25, P16, P11, P1); 2 studies use BPMN (P38, P41); and 5 studies reported the creation or use of UML Profiles (P29, P31, P44, P49, P52). For example, SoaML is an OMG graphical modeling language for designing systems according to service-oriented architecture (SOA), and SOAML4IOT is an extension of specific concepts for the IoT domain. Both languages are defined as UML profiles as discussed in study P29. Study P43 uses SysML to explore the application of model-based design. Study P60 describes SysML4IoT, a SysML profile based on the IoT-A reference model, and discusses model-based transformations from SysML into the NuSMV language. Study P10 describes the CHESSIoT framework that uses the CHESSML language, a UML profile also based on SysML and MARTE languages and implemented with the Papyrus workbench.

Study P38 applies the SEMIoTICS framework that directs using various modeling languages (for instance, Situation Modeling Language) and IoT platform technologies to enhance interoperability inside and between early warning systems. From a different perspective, Study P54 describes the OMG IFML (Interaction Flow Modeling Language) to support the design of software applications’ user interfaces.

Table 2 summarizes the modeling languages used in the reviewed papers.


**What tools are used? (RQ1.2)**


A modeling language has multiple facets, such as its abstract syntax (i.e., its concepts, relationships, and constraints) and concrete syntax (i.e., its textual or graphical concrete representation). These facets are implemented and supported by specific tools commonly known as language workbenches or language frameworks [13,14,39]. Popular language workbenches that provide graphical editors are DiaGen, Eugenia, GMF, Graphiti, MetaEdit+, Obeo Designer, and Sirius [40], while those that support textual languages are Xtext and MetaEdit. Six (6) studies (P1, P4, P21, P32, P46, P56) refer to the use of textual languages implemented with the Xtext framework, while, regarding visual modeling languages (i.e., with graphical concrete syntax), 7 studies reported the use of Sirius (P3, P10, P14, P12, P22, P24, P35), 4 studies the use of Eugenia (P5, P7, P13, P50), 6 studies the use of GMF (P2, P7, P13, P65, P67, P50), and other studies report the use of language workbenches like Obeo Designer (P12) and MetaEdit+ (P39).

Table 3 presents the studies that refer to the implementation or use of textual and graphical modeling languages.


**What model transformations and the respective supported languages and tools are employed? (RQ1.3)**


Model transformations are used in MDE to convert models from one representation to another. There are various languages and tools that have been used to perform these transformations. For instance, Meta3, Xpand, and Xtend covered both M2M and M2T transformation, while Acceleo and JET covered only M2T transformation. However, despite their popularity, some of these frameworks, like the JET (Java Emitter Templates) are not maintained anymore.

Twenty-five (25) studies refer to using some transformations. Eighteen (18) studies refer using M2T (P1, P5, P6, P7, P10, P11, P13, P14, P22, P23, P24, P25, P26, P27, P36, P47, P62, P68) and 14 studies report using M2M transformations (P5,P6, P11, P15, P19, P23, P26, P27, P28, P41, P52, P58, P62, P68). Only 8 studies refer to using both M2T and M2M transformations. Eleven (11) studies refer the implementation of M2T transformations with Accelo (P5, P6, P10, P13, P14, P19, P22, P24, P36, P62, P67) and 7 studies refer M2M transformation with the ATL framework (P5, P6, P26, P41, P42, P52, P62).

Table 4 shows the distribution of papers by types of model-based transformations and the supported tools.


**What are the outputs of M2M and M2T transformations? (RQ1.4)**


M2M and M2T transformations are techniques used in MDE to convert models from one representation to another. The output of an M2M transformation is a new model in a different format or representation. This new model is typically semantically equivalent to the original but may have a different structure or syntax. On the other hand, the output of an M2T transformation is text, typically in the form of source code, but it also can be in the form of documentation, configuration files, or other textual artifacts. The text generated is typically based on the structure and/or the information contained in the original model.

In general, M2M transformations are used to transform models from one meta-model to another, for example, from UML to SysML, or from one notation to another, for example, from BPMN to EPC. In comparison, M2T transformations generate source code, documentation, or other textual artifacts from a model. Both M2M and M2T transformations can be performed using a variety of languages and tools, such as QVT, ATL, Acceleo, and Xtend.

For example, Study P10 includes an advanced multi-platform code generation framework that supports multiple target programming languages, such as C, C++, Java, Arduino, and JavaScript. Study P11 uses the ATL framework and discusses M2M and M2T transformations of platform-specific models and software code for systems like RIOT, Arduino, Contiki, and TinyOS, with code written for multiple programming languages C, nesC, Java, JSON, XML. Study P14 discusses a semi-automatic process for generating Ballerina code for RESTful and orchestrator services, Arduino code for deploying IoT nodes, NCL-Lua code for the DTV interface, and Android code for smartphones. Study P24 reports using Acceleo to generate code for programming languages like Flutter, React JS, and VHDL. Studies P8, P56, P59, and P67 discuss M2T transformations with the generation of C++ or Java code. Study P62 discusses M2T transformations that generate Java code and SQL DDL scripts from JavaSE and database models.

### 5.2. What Application Domains Are These MDE Approaches Applied to? (RQ2)

MDE approaches and tools are applied to a wide range of application domains in IoT, including for Smart Home Automation, Industrial Internet of Things, Smart Grid, Healthcare, Agriculture, Predictive maintenance [41].

However, from the selected papers the most popular domains are the followings: smart buildings and living, smart healthcare, smart environment, smart cities, smart energy, smart transport, and mobility. Table 5 shows the distribution of papers by application domains.

Eight (8) studies show or discuss MDE4IoT techniques applied into the Smart Manufacturing domain (P4, P7, P10, P23, P24, P51, P53, P30); 10 studies into Smart healthcare domain (P4, P7, P10, P23, P24, P51, P53, P30); 3 studies into Smart Cities (P48, P61, P16); 5 studies into Smart Environment (P19, P35, P45, P62, P50); 2 studies into Smart Energy (P1, P39); 16 studies into Smart buildings and living (P1, P5, P7, P8, P13, P14, P18, P20, P21, P22, P27, P28, P32, P36, P47, P53); 6 studies into Smart Transportation (P6, P26, P37, P38, P44, P64); and 3 studies into Smart Agricultural (P11, P43, P25).

*Smart Manufacturing* aims to increase factories’ efficiency by enhancing manufacturing, process control, and monitoring systems. *Smart healthcare* aims to enhance medical systems, making them more efficient, convenient, and personalized, by supporting remote patient monitoring, telemedicine, and medical device management systems [42]. *Smart cities* is a new domain that not only improves the quality of life in a city but also lowers the operating costs of public services like transportation, parking, and waste management systems [43]. *Smart Environment* involves rare event detection, is in high demand because it can warn against natural disasters; may also monitor various environmental factors such as air quality, noise level, and weather conditions, to improve the overall quality of life in cities and other areas. *Smart Energy* refers to the improvements in the distribution and consumption of utilities such as electricity, gas, and water, and may involve control and monitoring lighting in buildings, streets, and other public areas, to increase energy efficiency and reduce costs [41]. *Smart buildings and living* aims to make buildings more energy-efficient and enhance the quality of life of their users [41]. *Smart transport* is a method of incorporating modern technologies into transportation systems. Cloud computing, wireless communication, location-based services, computer vision, and other mobility-enhancing tools are all part of this [44]. *Smart Agricultural* refers to systems that may monitor and control various aspects of agriculture, such as irrigation, fertilization, and crop management, to increase efficiency, reduce environmental impact and increase crop yield [41].

### 5.3. What DAML Techniques Are Supported by These MDE Approaches? (RQ3)

DAML methods and techniques are crucial for developing smart IoT services and CPS applications [45]. After analyzing these studies, we identified only four studies that discussed the application of DAML techniques in the scope of MDE4IoT: P1, P4, P33, and P15.

Study P1 introduced the ML-Quadrat (ML2) framework that supports supervised ML with different techniques such as Logistic Regression for linear classification, Linear Regression, Gaussian Naive Bayes, Multinomial Naive Bayes, Complement Naive Bayes, Bernoulli Naive Bayes, Categorical Naive Bayes, Decision tree Regressor and Decision Tree Classifier, the random forest regressor and Random Forest Classifier ensemble methods, The Multi-Layer Perceptron (MLP) Artificial Neural Networks (ANN). ML2 also supports unsupervised ML-included techniques, like K-Means, Mini-Batch K-Means, DB-SCAN, spectral clustering, and the Gaussian Mixture Model, and semi-supervised ML support techniques like self-training, label propagation, and label spreading (See Table 6).

Study P4 proposes a novel approach to software and AI engineering for smart IoT services that allow data analytics and machine learning components to run in part or entirely on IoT edge devices, which may be severely limited in terms of power and computational resources. The proposed approach for software development is based on the DSM methodology of the MDSE paradigm. In particular, it builds on the previous work of ML-Quadrat [P1].

Study P15 proposed the MoSIoT framework, which integrates a prediction module into the MoSIoT framework that applies an integrated set of learning attributes to the scenario model related to ML algorithms. This framework applied an approach based on GreyCat, which seamlessly integrated machine-learning algorithms into a domain model.

Study P33 integrated ML algorithms with software models. This work implemented the following algorithms: (1) Regression: Live linear regression; (2) Classification: Live decision trees, Naive Bayesian models, Gaussian Bayesian models; and (3) Profiling: Gaussian Mixture Models (Simple Multinomial).

Due to the significance of time series forecasting for IoT systems, we examined these studies to determine whether they discussed or proposed time series modeling techniques. However, our analysis shows that none of the studies has yet considered time series modeling techniques.

## 6. Discussion

In addition to the thorough evaluation of the outcomes of the scoping review provided in Section 5, this section includes a general discussion of the acquired findings and their consequences.

### 6.1. Main Findings

Regarding modeling languages addressed by RQ1, results from RQ1.1 showed that the top 3 most used modeling languages for IoT applications are: DSLls, UML, and UMLProfile. The most popular type of modeling language is DSLs which is reported in 29 studies. Then, UML with 11 studies, and UML Profile with 5 studies reported.

Concerning question RQ1.2, our analysis revealed that 6 studies used Xtext and MontiCore with 1 study reported for textual modeling language tools. Regarding graphical modeling language tools, 7 studies reported using Sirius, 6 using GMF, 4 using Eugenia, 1 using Metaedit+, and 1 using Obeo-Designer. These findings showed that graphical modeling language makes it easy to understand because most people benefit from a visual aid.

The analysis of the results from RQ1.3 showed that the M2T transformation, with 18 studies, gains more attention in terms of the existing or developed tools and languages compared to the M2M transformation with only 14 studies. In addition, it is observed that tools like Accelo and ATL are the most popular for model-based transformations, being Accelo the most tool for M2T transformation, as reported in 10 studies.

For the application domain, RQ2 results revealed that most studies reported IoT applications in the smart buildings and living application domains, which were mentioned in 16 studies out of 68. The other application domains were smart healthcare (with 10 studies), smart environment (with 5 studies), smart cities (with 3 studies), smart manufacturing (with 8 studies), smart transport and mobility (with 6 studies), smart agricultural and smart energy (with 2 studies).

Regarding question RQ3, concerning the use of DAML techniques in MDE4IoT, We find 4 studies among all of the analyzed studies. Few researchers have developed supervised machine learning techniques (reported in only 2 studies), unsupervised machine learning (4 studies), and semi-supervised machine learning (4 studies) for this purpose. We also verify that none of the studies supports time series forecasting. The results show that the total number of studies that reported using DAML is quite low (only 4 studies out of 68) for a wide and complex domain like the IoT.

### 6.2. Study Limitations

The potential of missing primary research frequently threatens any literature review. The databases we checked might not have included all the pertinent studies. However, to increase external validity, we employed several strategies. First, we looked at the author profiles of the most prolific authors to see if we missed any relevant studies. Second, because we were all familiar with the subject, we talked about whether we were aware of any studies that might have been relevant but were suspiciously missing from the sample. Third, we looked at the first 200 Google Scholar results from the same search.

However, despite these safeguards, it is possible that some pertinent studies were still overlooked. The results are biased toward publishing because we decided not to include unpublished work. Still, based on this work, we can observe some research issues, which are mentioned in the next section as future work.

## 7. Conclusions

MDE approaches may positively affect the quality of IoT systems easily by managing the increasing complexity of such IoT systems. MDE helps easily integrate various technologies (or migrate to new technologies) and make them compatible with the required changes, as well as the maintenance and documentation of the facilities.

The scoping review presented in this paper aimed to analyze and discuss the current state of the art of the MDE applied to the IoT domain. To do this, we selected 68 papers in the field, published between 2010 and 2022, and focused on languages, tools, application domains, and data analytics and machine learning techniques in the MDE field, specifically related to the IoT domain.

Our findings indicate that DSLs were favored above other modeling languages. Xtext and Sirius are the most popular tools for both textual and graphical modeling languages, respectively. Regarding the application domains, smart buildings and living are the most commonly used domains for IoT applications, namely, in what concerns the studies analyzed. Additionally, our review found that few studies have proposed MDE approaches for DAML techniques, and none have considered techniques for analyzing time series data.

Based on our review, it is clear that there is a need for further research in the area of MDE for IoT. Specifically, studies need to investigate the use of different types of modeling languages and tools for IoT systems and explore the various application domains where MDE can be applied.

Additionally, research is needed to explore integrating DAML techniques with MDE for IoT, specifically concerning time series data. This could include developing new methods and techniques for effectively representing and managing the complexity of time series data in the MDE process and for integrating DAML techniques into the MDE process to allow for efficient and effective analysis and processing of time series data. Time series data can be highly dynamic and involve many variables, making creating accurate and comprehensive models difficult. Another challenge is the need to integrate DAML techniques into the MDE process in a way that allows for the efficient and effective analysis and processing of time series data. This may require developing specialized methods and approaches for working with DAML in MDE for IoT services. A further challenge is the need to ensure the scalability and performance of MDE for IoT services when working with large amounts of time series data. This may involve specialized techniques and tools for optimizing the MDE process in this context.

## Figures and Tables

**Figure 1 sensors-23-01458-f001:**
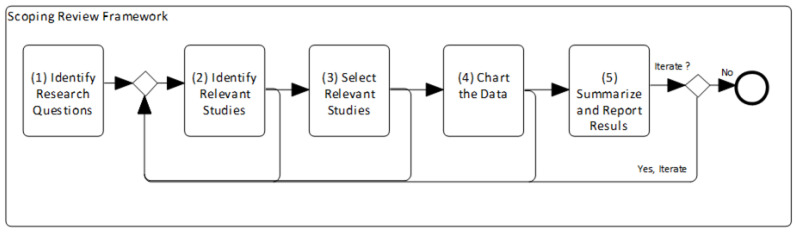
Followed stages of the scoping review framework.

**Figure 2 sensors-23-01458-f002:**
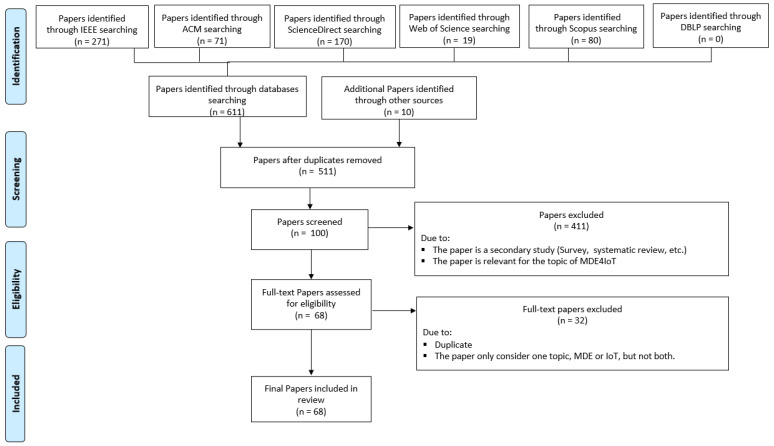
PRISMA flow diagram for paper selection.

**Figure 3 sensors-23-01458-f003:**
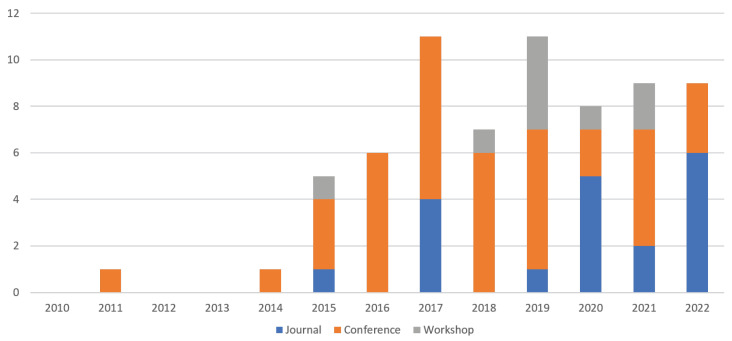
Distribution of papers by year and publication type.

**Figure 4 sensors-23-01458-f004:**
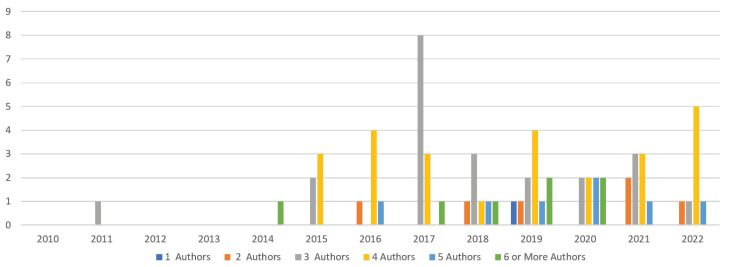
Distribution of papers by year and the number of authors.

**Figure 5 sensors-23-01458-f005:**
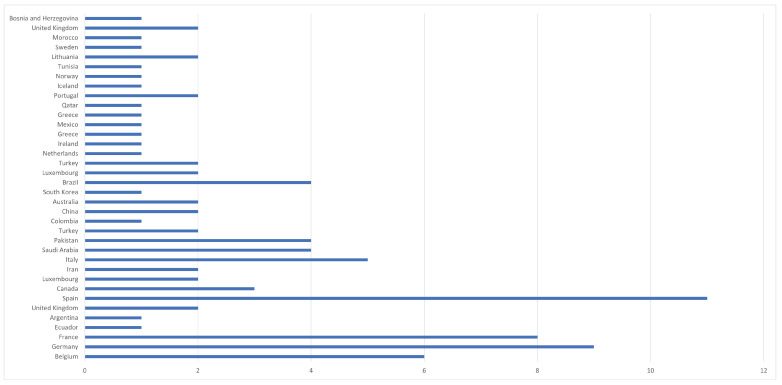
Distribution of papers by country.

**Figure 6 sensors-23-01458-f006:**
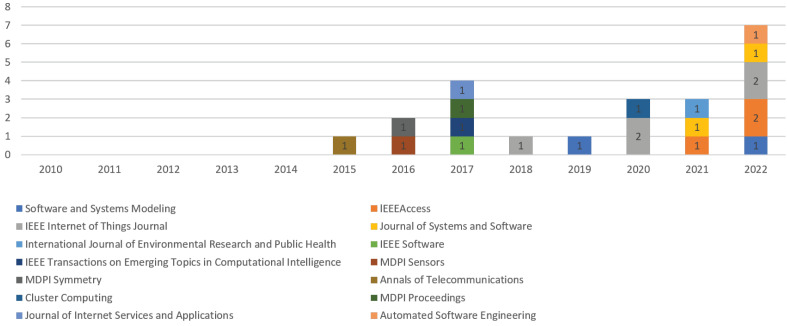
Distribution of papers by year and Journal.

**Figure 7 sensors-23-01458-f007:**
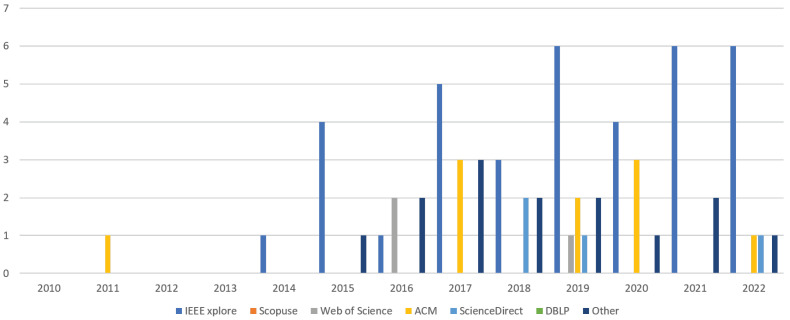
Distribution of papers by year and digital databases.

**Table 1 sensors-23-01458-t001:** Related works.

Authors	Study	Paradigm	Target System	Analyzed Papers	Years
Sabin et al., 2018	[27]	MDE	IoT	26	N/A
Felicien et al., 2020	[28]	Low-code Engineering	IoT	16	2000–2020
Muzaffar et al., 2017	[29]	MDE	Cloud computing	25	2009–2016
Abshir Mohamed et al., 2021	[30]	MDE	CPS	140	2010–2018
Casalaro et al., 2022	[31]	MDE	Mobile Robot systems	69	2000–2022
Mashkoor et al., 2021	[32]	MDE	Safety and Security Systems	95	1992–2020
Salman et al., 2020	[15]	DSL	IoT	23	2014–2020
Edsonde et al., 2021	[33]	MDE	Robotics	63	2014–2022

**Table 2 sensors-23-01458-t002:** Distribution of papers by modeling language.

PID	Framework	DSLs	UML	BPMN	SysMl	UML Profile	Other
P1	ML-Quadrat (ML2)	✓	✓				
P3	Monitor-IoT	✓					
P5	SimulateIoT-FIWARE	✓					
P6	CyprIoT-DSL	✓					
P7		✓					
P9	HealMA	✓					
P10	CHESSIoT						✓
P11	FTG+PM	✓	✓				
P13	SimulateIoT	✓					
P14		✓					
P16	FaultFlow		✓				
P18		✓					
P19	DSML4contiki	✓					
P20		✓					
P22		✓					
P24		✓					
P25			✓				
P29	SoaML4IoT	✓				✓	✓
P30		✓					✓
P31						✓	
P32	Cypriot	✓					
P35		✓					
P36		✓					
P37			✓				
P38	Semiotics		✓	✓			✓
P39	SiMoNa	✓					
P41				✓			
P42			✓				
P43					✓		
P44	UMLOA					✓	
P46		✓					
P47	SmartHomeML	✓					
P48		✓					
P49	UML4IoT					✓	
P50		✓					
P51	ThingML		✓				
P52	IoTA-MD					✓	
P53			✓				
P54							✓
P55			✓				
P56		✓					
P59		✓					
P60	SysML4IoT				✓		
P61	MDE4IoT		✓				
P65	FRASAD	✓					
P66		✓					
P68		✓					
Total		29	11	2	2	5	5

**Table 3 sensors-23-01458-t003:** Distribution of papers by textual and graphical modeling language tools.

		Textual Modeling Language Tools		Graphical Modeling Language Tools				
PID	Framework	Xtext	MontiCore	Sirius	Obeo Designer	GMF	Metaedit+	Eugenia
P1	ML-Quadrat (ML2)	✓						
P2						✓		
P3	Monitor-IoT			✓				
P4		✓						
p5	SimulateIoT-FIWARE							✓
P6	CyprIoT-DSL							
P7						✓		✓
P9	HealMA			✓				
P10	CHESSIoT							
P12				✓	✓			
P13	SimulateIoT					✓		✓
P14				✓				
P21	IoTSuite	✓						
P22				✓				
P24				✓				
P30			✓					
P32	Cypiot	✓						
P35				✓				
P39	SiMoNa						✓	
P46	EL4IoT	✓						
P50						✓		✓
P56		✓						
P65	FRASAD					✓		
P67						✓		
Total		6	1	7	1	6	1	4

**Table 4 sensors-23-01458-t004:** Distribution of papers by types of model-based transformations and tools.

		Transformation		Tools for Transformation			
PID	Framework	M2T	M2M	Accelo	ATL	M2T Output	M2M Output
P1	ML-Quadrat	✓				Java, Python	
P2						Intermediary Macro-Code Generation, Code Generation for Node-RED Target, Java code application	
P3							
P4		✓				C for Arduino, Python	
P5	SimulateIoT-FIWARE	✓	✓	✓	✓		
P6	CyprIoT-DSL	✓	✓	✓	✓	C, Java, Arduino, AC rules and documentation	
P7		✓				Java, configuration code	
P8	MontiThings					C++	
P9	HealMA	✓		✓		Java, XML	
P11	FTG+PM	✓	✓			C, nesC, Java, JSON, XML, GenerateContiki Code, GenerateTinyOSCode, GenerateJavaCode, GeneratePetrinetConfiguration, GenerateArduinoCode, GenerateRIOTCode	RIOTMode, SystemModel, GenerateContikiModel, GenerateTinyOSModel, GenrateGatewayModel, GeneratePetrinetModel, GenerateNodeRedModel, GenerateArduinoModel, GenerateRIOTModel, GenerateRIOTModel
P13	SimulateIoT	✓		✓		Java, configuration code	
P14		✓		✓		Java, Xml, Ardunio code, json RESTful APIs Associated	
P15	MoSIoT		✓				
P19	DSMl4Contiki		✓	✓			Petri-net Models
P22		✓		✓		Arduino code files (.ino), code Node-Red(.json), Ballerina code (.bal)	
P23	AutoIoT	✓	✓				
P24		✓		✓		Flutter, React JS, And VHDL.	
P25		✓					
P26		✓	✓		✓		
P27		✓	✓				
P28	BRAIN-IoT		✓			Java, Osgi Artifact, C	
P33	GreyCat					Java, TypeScript.	
P35						Code for Mote (SourceNode), SinkNode, Raspberry Pi (Java Code), ESP8266 (Arduino code), configuration for IoT Log Manager	Petri-net Models
P36		✓		✓		Ardunio code	
P41			✓		✓		
P42					✓		
P47	SmartHomeML	✓					
P51	ThingML					C/C++, Java, and JavaScript and several libraries and open platforms (Arduino, Raspberry Pi, Intel Edison, Linux, and so on).	
P52			✓		✓		
P56						Java	
P58			✓				
P59						Java	
P61	MDE4IoT					C	
P62			✓	✓	✓	java code and SQL DDL	
P67				✓		Java	
P68		✓	✓				
Total		18	14	11	7		

**Table 5 sensors-23-01458-t005:** Distribution of papers by application domains.

PID	Framework	Healthcare	Agricultural	City	Energy	Manufacturing	Building	Environment	Transport
P1	ML-Quadrat (ML2)				✓		✓		
P4						✓			
P5	SimulateIoT-FIWARE						✓		
P6	CyprIoT-DSL						✓		✓
P7						✓	✓		
P8							✓		
P9	HealMA	✓							
P10	CHESSIoT					✓			
P11	FTG+PM		✓						
P13	SimulateIoT		✓				✓		
P14							✓		
P15	MoSIoT	✓							
P16	FaultFlow			✓					
P19	DSMl4Contiki							✓	
P20							✓		
P21							✓		
P22							✓		
P23	AutoIoT					✓			
P24						✓			
P25			✓						
P26									✓
P27							✓		
P28	BRAIN-IoT						✓		
P30						✓			
P31		✓							
P32	Cypriot						✓		
P35								✓	
P36							✓		
P37									✓
P38	Semiotics								✓
P39	SiMoNa				✓				
P40						✓			
P43		✓							
P44	UMLOA								✓
P45								✓	
P47	SmartHomeML						✓		
P48				✓					
P50								✓	
P51	ThingML	✓				✓			
P52	IoTA-MD	✓							
P53							✓		
P54		✓							
P55		✓							
P57		✓							
P58		✓							
P61	MDE4IoT			✓					
P62								✓	
P64									✓
Total		10	3	3	2	8	16	5	6

**Table 6 sensors-23-01458-t006:** Data Analytics and Machine Learning in MDE4IoT.

		Types of Machine Learning Algorithms			
PID	Framework	Supervised	Unsupervised	Semi Supervised	Time Series Models
P1	ML-Quadrat(ML2)	Logistic Regression Linear Regression, Gaussian Naive Bayes, Multinomial Naive Bayes, Complement Naive Bayes, Bernoulli Naive Bayes, Categorical Naive Bayes, Decision Tree Regressor Decision Tree Classifier, Decision Tree Regressor Decision Tree Classifier, Random Forest Classifier, Multi-Layer Perceptron (MLP) Artificial Neural Networks (ANN)	K-Means, Mini-Batch K-Means, DB-SCAN, Spectral Clustering, Gaussian Mixture Model,	Self-Training, Label Propagation, Scikit-Learn Label Spreading	NO
P4		Logistic Regression Linear Regression, Gaussian Naive Bayes, Multinomial Naive Bayes, Complement Naive Bayes, Bernoulli Naive Bayes, Categorical Naive Bayes, Decision Tree Regressor Decision Tree Classifier, Decision Tree Regressor Decision Tree Classifier, Random Forest Classifier, Multi-Layer Perceptron (MLP) Artificial Neural Networks (ANN)	K-Means, Mini-Batch K-Means, DB-SCAN, Spectral Clustering, Gaussian Mixture Model,	Self-Training, Label Propagation, Scikit-Learn Label Spreading	NO
P15	MoSIoT		Live linear regression, Live decision trees Naive Bayesian models Gaussian Bayesian models baz	KNN, StreamKM++ Gaussian Mixture Models	NO
P33	GreyCat		Live linear regression, Live decision trees Naive Bayesian models Gaussian Bayesian models baz	KNN, StreamKM++ Gaussian Mixture Models	NO

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
