# Peer review of "Model-Driven Engineering Techniques and Tools for Machine Learning-Enabled IoT Applications: A Scoping Review"

_sensors, 2023, doi:10.3390/s23031458_

Round 1

Reviewer 1 Report

The article is a systematic literature review paper that targeted Model-Driven Engineering (MDE) tools and Domain Specific Languages (DSLs) for the Internet of Things (IoT) domain. 

The introduction section is very short. Authors must write the details of IoT, MDE tools, and DLS. The reason for this SLR, the rationale of this SLR, the justification that the existing literature review is not quite sufficient, and proof from the literature that the questions you discussed in this paper were never discussed before. 

The phases and activities diagram and its details are missing which is quite essential for the SLR. 

The discussion on the results displayed in Figure 2 is missing.

There is no need for Figure 3 in the presence of Table 3. Similarly Figure 4, 5, and 6.

I am not satisfied with the answers to the questions explored in this study. The questions that are asked in this study are not as novel. Authors are required to target those questions that are more helpful for the research community. 

Limitations of the study and future direction are missing in the conclusion section.

Authors are required to check each reference as various references have missing details.   

Author Response

Please, see attached submitted PDF.

Reviewer 2 Report

The key contributions of this study as claimed by the authors are:

"This paper reviews the literature on Model-Driven Engineering (MDE) tools and Domain-Specific Languages (DSLs) for the Internet of Things (IoT) domain. In this paper, the authors study and classify prior work in the literature. The goal is to provide a survey of the state of art in MDE4IoT in the context of smart (i.e., DAML-enabled) services."

The writing quality of this work is good. However, the organization and presentation quality must be improved. There are some major concerns, such as:

1. The Abstract is good. However, the Conclusion must be revised and make it precise.

2. The reference literature in the Introduction section is insufficient. The authors must add more references. 

3. Section 2 is supported with good discussion. What's the motivation to add Background when you have a similar section of related works?

4. The authors must revise Table 1 and add more relevant references.

5. Research methodology is properly discussed. 

6. Table 3-7 are nicely organized. 

7. Section 4 is comprehensively discussed.

8. Section 6 is also supported with good discussion.

Author Response

Please, see attached submitted PDF.

Round 2

Reviewer 1 Report

After a detailed reading of the paper, I have come to the conclusion that the paper still needs major changes. 

The authors have no solid grip on the SLR. 

The authors have adopted the SLR method that is specified in reference No. 34.  The authors skipped many essential stages of the SLR like

In Stage 1:

1.1 Identify the needs of SLR 

1.2 Determine general and specific objectives

1.3 Identify the initial research questions

1.4 Develop Review Protocol

1.5 Evaluate Review Protocol

1.6 Identify relevant bibliographical databases

1.7 Commissioning a review

Similarly, stage 2 and onward stages are required detailed investigation. 

Sensors Journal is a high-impact factor journal and readers are supposed high-quality papers in this journal. 

Add distribution of the articles according to the:

Journal Name,

Publication Years,

Countries,

Sectors,

Type of research,

Data collection tools, and 

authorship pattern.

Stage 5 is poorly written. Please expend it.

The answers to the research questions are also not satisfactory. Authors are required to study the extracted 68 papers thoroughly before writing the answers to the research questions.

11 studies [P61, P55, P53, P51, P42, P38, P37, 312 P25, P16, P11, P1] reported the use of UML. Please cite papers instead of giving paper no. p61....

Appendix-A is irrelevant to include in the article. It is just added to increase the size of the paper. 

Authors are advised to read SLR papers published in reputed and high-impact factor journals before designing the methodology. 

I hope, the authors will improve their paper. 

Author Response

See attached PDF.
